

# Computer-assisted initial diagnosis of rare diseases

Rui Alves[1,2,*], Marc Piñol[3,*], Jordi Vilaplana[3,4], Ivan Teixidó[3,4], Joaquim Cruz[1,2], Jorge Comas[1,2,3,4], Ester Vilaprinyo[1,2], Albert Sorribas[1,2] and Francesc Solsona[3,4]

[1] Departament de Cienciès Mèdiques Bàsiques, Universitat de Lleida, Lleida, Catalunya, Spain
[2] IRBLleida, Lleida, Catalunya, Spain
[3] Departament d'Informàtica i Enginyeria Industrial, Universitat de Lleida, Lleida, Catalunya, Spain
[4] INSPIRES, Lleida, Catalunya, Spain
[*] These authors contributed equally to this work.

## ABSTRACT

**Introduction.** Most documented rare diseases have genetic origin. Because of their low individual frequency, an initial diagnosis based on phenotypic symptoms is not always easy, as practitioners might never have been exposed to patients suffering from the relevant disease. It is thus important to develop tools that facilitate symptom-based initial diagnosis of rare diseases by clinicians. In this work we aimed at developing a computational approach to aid in that initial diagnosis. We also aimed at implementing this approach in a user friendly web prototype. We call this tool Rare Disease Discovery. Finally, we also aimed at testing the performance of the prototype.

**Methods.** Rare Disease Discovery uses the publicly available ORPHANET data set of association between rare diseases and their symptoms to automatically predict the most likely rare diseases based on a patient's symptoms. We apply the method to retrospectively diagnose a cohort of 187 rare disease patients with confirmed diagnosis. Subsequently we test the precision, sensitivity, and global performance of the system under different scenarios by running large scale Monte Carlo simulations. All settings account for situations where absent and/or unrelated symptoms are considered in the diagnosis.

**Results.** We find that this expert system has high diagnostic precision ($\geq$80%) and sensitivity ($\geq$99%), and is robust to both absent and unrelated symptoms.

**Discussion.** The Rare Disease Discovery prediction engine appears to provide a fast and robust method for initial assisted differential diagnosis of rare diseases. We coupled this engine with a user-friendly web interface and it can be freely accessed at http://disease-discovery.udl.cat/. The code and most current database for the whole project can be downloaded from https://github.com/Wrrzag/DiseaseDiscovery/tree/no_classifiers.

# INTRODUCTION

A rare or orphan disease affects a small fraction of the population. This fraction is less than 200,000 individuals in the total population in the USA, less than 50,000 individuals in Japan, and less than 2,000 in Australia. In Europe, diseases are rare if they affects less than

---

Corresponding authors
Rui Alves, ralves@cmb.udl.cat
Francesc Solsona,
francesc@diei.udl.cat

one in every 2,000 individuals (*EURORDIS Consortium*, *2016*; *Lavandeira*, *2002*; *Schieppati et al.*, *2008*). Overall, more than 10,000 such diseases have been documented (*Rath et al.*, *2012*; *McKusick*, *2008*; *ORPHANET*, *2015*), and about 10% of the population suffers from some rare disease (*Schieppati et al.*, *2008*). Most known rare diseases have genetic origin (*Rath et al.*, *2012*; *McKusick*, *2008*). The association between specific diseases and the genes that might cause them can be found at the OMIM database (*McKusick*, *2008*).

Because of their low individual frequency, initial diagnosis of rare diseases by clinicians is not always easy (*Polizzi et al.*, *2014*). Often, those clinicians might never have been exposed to patients suffering from the disease. In addition, as it can be seen in ORPHANET (*Maiella et al.*, *2013*), many different diseases have a partially overlapping spectrum of symptoms that can confuse the diagnosis. In general, conclusive diagnosis for most rare diseases comes from a genetic test that identifies the genetic variations associated to that disease. These tests tend to be expensive and/or target a specific (small set of) disease(s). Given all these constraints, it is important to develop methods and tools to facilitate a quick and accurate symptom-based initial diagnosis of rare diseases.

Symptom-based diagnosis is a pattern recognition/classification problem, where an accurate prediction (the correct disease) must be made, based on a set of input characteristics (the symptoms). This is a classical computational problem, and computer-assisted medical diagnosis (CAD) can have many forms (*Eadie, Taylor & Gibson*, *2012*). CAD is routinely used in clinical image analysis (see for example *Wang & Summers*, *2012*), although other applications, such as telemedicine, are also becoming frequent (*Lopman et al.*, *2006*; *Soyer et al.*, *2005*; *Steele et al.*, *2005*).

Symptom-based Differential Diagnosis (DDX) generators that assist medical doctors in automatically generating initial diagnosis have been originally developed in the mid nineteen eighties (*Barnett et al.*, *1987*). A recent comparative analysis of these methods reveals that their accuracy and sensitivity did not significantly improve since (*Bond et al.*, *2012*; *Umscheid & Hanson*, *2012*). Nevertheless, DDX generators could in principle be an optimal solution for an initial diagnosis of rare diseases. However, given that the cost-effectiveness of eHealth solutions appears to be debatable (*Black et al.*, *2011*; *Free et al.*, *2013a*; *Free et al.*, *2013b*; *McLean et al.*, *2013*; *Eysenbach et al.*, *2002*; *Avery et al.*, *2012*; *Free et al.*, *2011*; *Howitt et al.*, *2012*; *Sheikh et al.*, *2014*; *Morrison et al.*, *2013*; *Huckvale et al.*, *2012*; *Greenhalgh & Swinglehurst*, *2011*; *Mazzucato, Houyez & Facchin*, *2014*), using such a computer-based solution should either be free or have a low cost.

Some DDX generators dedicated to assisting in the diagnosis of genetic diseases have been previously developed. They permit associating symptoms to diseases that can also be rare. Recent examples are Phenomizer (*Köhler et al.*, *2014*; *Köhler et al.*, *2009*), FindZebra (*Dragusin et al.*, *2013*; *Winther et al.*, *2014*), PhenoTips (*Girdea et al.*, *2013*), or Phenotip (*Porat et al.*, *2014*). While the first two are aimed at assisting in diagnosing any genetic disease from a list of symptoms provided by the user, the latter two have a somewhat different purpose. PhenoTips mostly provides a framework to share and analyze patient data between professionals. Once data is introduced, it can also be used for assisted diagnosis. In contrast, Phenotip focuses on prenatal diagnosis, which limits the symptoms it uses to those that can be obtained from pre-natal analysis methods. These four tools are

freely available to the medical community. Other free computational tools that assist in rare disease patient treatment and management do exist. For example, RAMEDIS (*Töpel et al.*, *2010*) provides a highly accurate and manually curated resource of human variations and corresponding phenotypes for rare metabolic diseases. DiseaseCard (*Lopes & Oliveira*, *2013*) provides a similar service, with automated curation. The office of Rare Diseases from the NIH (*Daneshvari, Youssof & Kroth*, *2013*; *Genetic and Rare Diseases (GARD) Information Center*, *2016*) is another useful resource for diagnosis and follow up of rare diseases. However, these tools do not provide a DDX generator that allow for doctors to get quick differential diagnostic options of rare diseases.

Developing a specific and freely accessible DDX generator for rare diseases requires two types of data. First, appropriate data sources associating specific symptoms to genetic diseases should be available. A highly curated, often updated, dataset containing information about the association between symptoms and rare diseases is available to the community at ORPHANET (*Maiella et al.*, *2013*). Second, a large and freely available golden standard dataset of rare disease patients is needed to test and validate the DDX generator. Although many initiatives are collecting data for tens to hundreds of thousands of rare disease patients (e.g., *Choquet & Landais*, *2014*; *Koutouzov*, *2010*), these dataset have yet to made publicly available.

With these constraints in mind we set out to develop and test a prototype rare disease DDX generator, which we call Rare Disease Discovery. Using the ORPHANET dataset as a source of information regarding the association between symptoms and rare diseases, we developed Rare Disease Discovery (RDD, http://disease-discovery.udl.cat/). This free DDX generator prototype is specific for rare diseases and automatically predicts the most likely rare diseases based on the known set of symptoms provided by the user.

## METHODS

### Data sources & software

A highly curated list of rare diseases, associated to their respective symptoms, was downloaded from ORPHANET (*ORPHANET*, *2015*) on September 2015. A MySQL database where each disease is associated to its symptoms was built. The web technology underlying RDD is described in detail in the Supporting Methods section of Appendix S1. All calculations and experiments were done using local Mathematica scripts.

### Diagnostic score function and disease ranking

The goal of RDD is to estimate which are the most likely rare diseases a patient might suffer from, based on the symptoms shown by that patient and on the symptoms that are associated to each rare disease in the ORPHANET dataset. To rank diseases and provide a differential diagnosis, RDD uses the scoring function $DS_i$ from Eq. (1).

$$DS_i = 1 - \frac{n}{\text{Max}[S_{\text{User}}, S_{\text{Disease } i}]}. \tag{1}$$

In Eq. (1), $S_{\text{User}}$ represents the number of symptoms provided by the user, $S_{\text{Disease } i}$ represents the number of symptoms of disease $i$ stored in the database, and

Max$[S_{\text{User}}, S_{\text{Disease } i}]$ represents the largest number between $S_{\text{User}}$ and $S_{\text{Disease } i}$. $n$ represents the number of symptoms that are different between the set submitted by the user and the set associated to any given rare disease in the database. The fraction $\frac{n}{\text{Max}[S_{\text{User}}, S_{\text{Disease } i}]}$ is always smaller than two and larger than or equal to zero. If all symptoms submitted by the user are the same as those from a disease and that disease has exactly the same symptoms as those provided by the user, $n = 0$. If the symptoms provided by the user and the symptoms associated to a disease are all different and $S_{\text{User}} = S_{\text{Disease } i}$, $n = 2$. Thus, $-1 \leq DS_i \leq 1$.

RDD differentially diagnoses a patient by letting the user choose the list of symptoms that are relevant for the specific case of interest. Once this list is selected, RDD calculates $DS_i$ for all diseases stored in the database. Then, RDD ranks the diseases in order of increasing $DS_i$, presenting the disease with the highest score as the most likely. In the Supporting Methods section of Appendix S1 we discuss the performance of other scoring functions and prediction methods that were tested and discarded.

### Retrospective study of previously diagnosed rare disease patients

We selected all usable patients with a confirmed rare disease diagnosis from the RAMEDIS (*Töpel et al.*, *2010*) collection of patients in order to retrospectively use their symptoms and evaluate the diagnostic performance of RDD on a real set of patients. See Supporting Methods and Supporting Figure 1 in Appendix S1 for selection details.

### Calculating sensitivity and accuracy of the DDX predictions and significance of the $DS_i$ score

Monte Carlo simulations were used to calculate the precision $p$ and sensitivity $s$ of RDD. $p$ is given by Eq. (2) and $s$ is given by Eq. (3):

$$p = \frac{\text{number of correctly predicted rare diseases}}{\text{number of correctly predicted rare diseases} + \text{number of incorrectly predicted rare diseases}} \quad (2)$$

$$s = \frac{\text{number of correctly predicted rare diseases}}{\text{total number of rare diseases}}. \quad (3)$$

A prediction is considered to be correct if the disease that is ranked by $DS_i$ as the most likely is the correct disease.

The global performance of RDD was also calculated using the F1-Score, which is the harmonic mean of $p$ and $s$:

$$\text{F1-Score} = 2\frac{p \times s}{p + s}. \quad (4)$$

In addition, Monte Carlo simulations were also used to calculate the statistical significance of the score $DS_i$. All simulations were done using Mathematica (*Wolfram*, *1999*).

### Benchmarking the Rare Disease Discovery algorithm

RDD's performance was benchmarked using four sets of experiments, all run using Stochastic Monte-Carlo simulations. These experiments are detailed in the Supporting Methods section of Appendix S1. The first experiment tested how a combination of

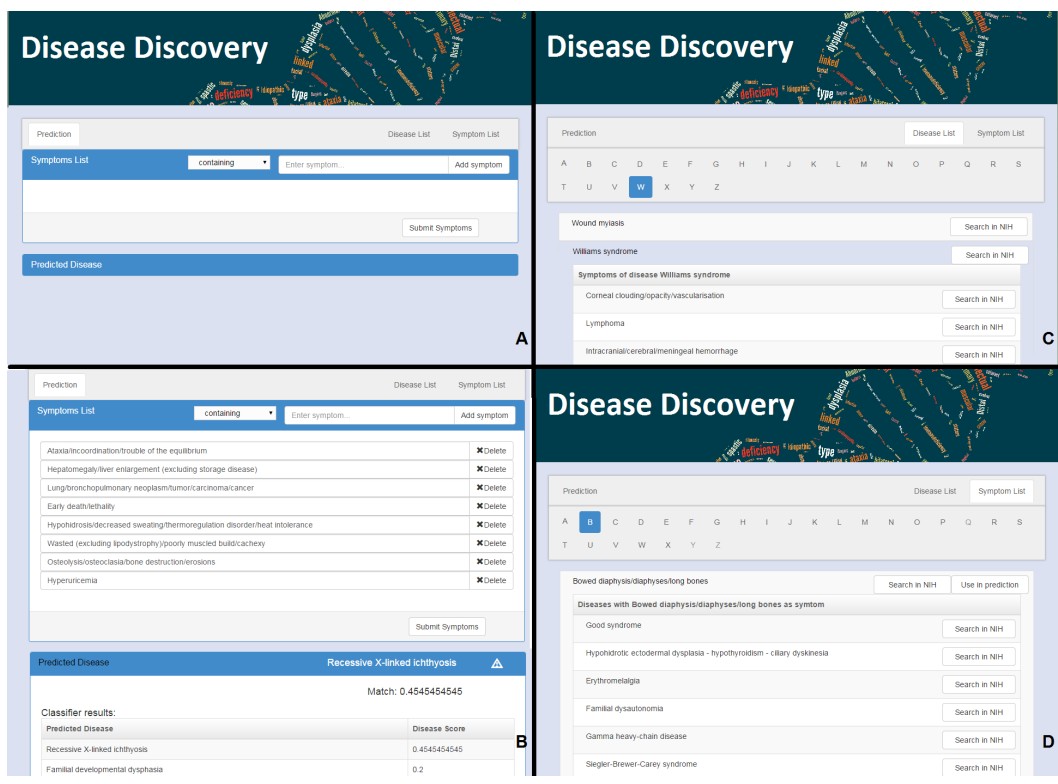

**Figure 1** **The web interface for Rare Disease Discovery.** (A) Entry screen. Users can type and select various symptoms. Once all relevant symptoms have been selected, the user can press ''Submit Symptoms.'' (B) Example of a differential diagnosis provided by the program. (C) List of diseases in the database, with its associated symptoms. (D) List of symptoms in the database, with its associated diseases.

unreported and unrelated symptoms affects prediction outcome of the RDD algorithm. The second experiment tested how unreported symptoms affect the prediction performance of the RDD algorithm. The third and fourth experiments estimate the minimum value for $DS_i$ that can be considered to be statistically significant and the minimum difference between $DS_i$ values that is statistically significant, respectively.

# RESULTS

## Using the Rare Disease Discovery web

RDD is available at http://disease-discovery.udl.cat/. A simple web interface is provided to the user (Fig. 1A). Users can search for individual symptoms by typing on the text field. Once the relevant symptom is identified, it must be selected. The user can type and select as many symptoms as required. Symptoms that are absent from the database are not accepted by the server. Once all relevant symptoms are selected, pressing the ''Submit Symptoms'' will generate a ranked list of disease predictions. The disease with the highest score is shown. If the score of the predicted disease is not statistically significant, this is indicated by a ⚠ symbol. Clicking the name of the disease will lead the user to the ORPHANET webpage where s/he can look up more information about the disease. In addition, when the user clicks the link ''Predicted Disease,'' RDD unfolds the full list of ranked diseases in

the browser (Fig. 1B). If the user clicks the name of any of the diseases, s/he will be taken to the RDD webpage with the list of symptoms for that disease. On that page s/he can also find external links about the disease at NIH.

The alphabetically ordered list of rare diseases, together with its associated symptoms can be directly accessed by pressing the "Disease List" pane (Fig. 1C). The alphabetically ordered list of symptoms, together with its associated diseases can be accessed by pressing the "Symptom List" pane (Fig. 1D). Users can also select symptoms directly from this pane and use them for prediction. NIH searches can be automatically launched for the disease or the symptom of interest.

### Illustrative examples of RDD usage

To illustrate the use of RDD, we randomly selected 10 diseases from the database. For each disease, we:

(1)   Calculate the number of symptoms associated to that disease in the database.
(2)   Randomly select the order of those symptoms.
(3)   Use the first symptom in the list to predict the disease.
(4)   Calculate the $DS_i$ score and rank for the disease.
(5)   Use the first five symptoms in the list to predict the disease (see Supporting Table 1 in Appendix S1).
(6)   Calculate the $DS_i$ score and rank for the disease.
(7)   Identify the minimal number of symptoms that rank the original disease as the most likely prediction, and the score associated to those symptoms.

Results are summarized in Table 1. We see that one symptom is not sufficient to correctly identify any of the tested rare disease as the most likely. However, with 5 symptoms, 7 of the diseases are correctly predicted (although with $DS_i$ scores below significance level), and 9 of the diseases are ranked among the top-2 most likely.

### Retrospective study of previously diagnosed rare disease patients

A small retrospective study was also performed to evaluate the performance of the RDD prototype. We obtained short report cards for 187 anonymous patients from the RAMEDIS (*Töpel et al.*, *2010*) database (see Supporting Methods and Supporting Figure 1 in Appendix S1 for details on report cards and patient selection). Symptoms were clearly itemized and described at best in 30% of the cards. In the remaining cases, descriptions could fit alternative symptoms. In such cases all alternative symptoms are considered. Given that 111 of the patients have three or less reported symptoms, errors are expected to be large. A clinician, with direct knowledge about the patient, is unlikely to introduce such errors in the diagnostic process. In spite of the noise in the data set, RDD included the correct disease in the list of predictions for 117 out of 187 patients. In 60% of these patients, the clinically diagnosed disease was on the top ten list of predictions (see text and Supporting Figure 2 in Appendix S1 for details). This percentage goes up to 80% if we consider the top fifty predictions for each patient. We note that only approximately 17% of the predictions had a score that was significant (>0.5).

**Table 1** **Examples of prediction results for a randomly chosen set of ten rare diseases.** Diseases are identified in column 1. Column 2 indicates the total number of symptoms associated to the disease in the ORPHANET dataset. Column 3 presents $DS_i$ for the disease when one symptom is submitted to RDD, as well as the ranking of the disease in the list of predictions. Column 4 shows $DS_i$ for the disease when five symptoms are simultaneously submitted to RDD, as well as the ranking of the disease in the list of predictions. Column 5 displays minimum $DS_i$ at which the disease is ranked as the most likely prediction, as well as the number of symptoms needed for that value of $DS_i$ to be obtained. Finally, column 6 indicates the number of symptoms that make $DS_i \geq 0.5$ for the disease, which is the value above which $DS_i$ is statistically significant. Details about the symptoms are given in Supporting Table 1 of Appendix S1.

| Disease | Number of associated symptoms | Score at 1 symptom (rank) | Minimun score at rank 1 (number of symptoms) | Number of symptoms for statistically significant score ($DS_i > 0.5$) |
|---|---|---|---|---|
| Beta-Thalassemia | 23 | 0.043(67th) | 0.13(3) | 12 |
| Canavan disease | 19 | 0.053(23rd) | 0.26(5) | 10 |
| Down syndrome | 48 | 0.021(244th) | 0.083(4) | 24 |
| Fabry disease | 66 | 0.015(111th) | 0.12(8) | 33 |
| Goldblatt syndrome | 23 | 0.043(81st) | 0.13(3) | 12 |
| Turner syndrome | 26 | 0.038(21st) | 0.077(2) | 13 |
| Uncombable hair syndrome | 7 | 0.14(1st) | 0.14(1) | 4 |
| Williams syndrome | 180 | 0.006(121st) | 0.028(5) | 90 |
| Yunis-Varon syndrome | 66 | 0.015(7th) | 0.14(9) | 33 |
| Zellweger-like syndrome without peroxisomal anomalies | 25 | 0.042(31st) | 0.12(3) | 13 |

## Comparison to other DDX generators

RDD's performance was also compared with that of other DDX engines that were freely available for illustrative purposes. After searching through the literature and the programs analyzed in Bond et al. (2012), this limited us to our own RDD (RareDiseaseDiscovery), in addition to DiagnosisPro, ISABEL, Phenomizer, and FindZebra. While DiagnosisPro and ISABEL are general DDX generator, RDD, FindZebra and Phenomizer are DDX generators that are specific for genetic diseases. We did not include the disease diagnostic assistance tool from ORPHANET in the comparison because that service is no longer maintained.

By using the same ten diseases with their associated symptoms described in columns two and four of Supporting Table 1 of Appendix S1, we asked each of the DDX generators to come up with a diagnosis of the disease. Results are summarized in Table 2. ISABEL identifies the correct disease as a possibility in three of the ten diseases. Diagnosis Pro identifies the correct disease as a possibility in four of the ten diseases. FindZebra identifies correctly nine of the diseases. RDD and Phenomizer identify the correct disease as a possibility in all ten cases.

Given that the examples from Table 2 were generated using symptoms from our database, there was the possibility that the performance of RDD was inflated with respect to the other DDX generators. To control for this we randomly selected ten patients from RAMEDIS for which RDD had included the correct diagnostic in the top ten list of predictions. Using the symptoms associated to each patient, we interrogated the five DDX engines using their default parameters and evaluated if the correct disease was diagnosed as a possibility results are summarized in Table 3. The performance of ISABEL and DiagnosisPro was significantly

**Table 2  Comparison of predictions between DDX generators.** Here we compare the most likely diagnosis of four well-known and freely available (at least for testing purposes) DDX generators with that provided by Rare Disease Discovery, when considering the joint symptoms used to perform the study summarized in columns 1 and 4 of Table 1.

| Disease | Diagnosis pro | ISABEL | Phenomizer | FindZebra | Rare Disease Discovery |
|---|---|---|---|---|---|
| Beta-Thalassemia | + | + | + | * | + |
| Canavan disease | * | * | + | + | + |
| Down syndrome | * | * | + | + | + |
| Fabry disease | + | + | + | + | + |
| Goldblatt syndrome | * | * | + | + | + |
| Turner syndrome | * | + | + | + | + |
| Uncombable hair syndrome | * | * | + | + | + |
| Williams syndrome | + | * | + | + | + |
| Yunis-Varon syndrome | * | * | + | + | + |
| Zellweger-like syndrome without peroxisomal anomalies | + | * | + | + | + |

Notes.
+ Suggests the appropriate disease in the top 10 ranked list of predictions.
* Does not suggest the appropriate disease in any position of the top 10 ranked list of predictions.

**Table 3  Comparison of predictions between DDX generators.** Here we compare the most likely diagnosis of four well-known and freely available (at least for testing purposes) DDX generators with that provided by Rare Disease Discovery. 10 patients with different symptoms and/or diseases were randomly selected from the RAMEDIS dataset. All symptoms were used.

| Disease (Patient ID) | Diagnosis pro | ISABEL | Phenomizer | FindZebra | Rare disease discovery |
|---|---|---|---|---|---|
| Classical homocystinuria (5) | + | ++ | ++ | ++ | ++ |
| Propionic acidemia (821) | + | + | + | * | ++ |
| Glycogen storage disease (1086) | + | ++ | + | ++ | ++ |
| Isovaleric acidemia (1050) | + | * | + | ++ | ++ |
| Galactosemia (970) | + | + | ++ | ++ | ++ |
| Carnitine palmitoyl transferase II deficiency (1024) | * | ++ | ++ | + | ++ |
| Canavan disease (492) | * | * | * | ++ | ++ |
| Porphyria (866) | + | * | ++ | ++ | ++ |
| Mitochondrial DNA depletion syndrome (940) | * | + | + | ++ | ++ |
| Congenital neuronal ceroid lipofuscinosis (830) | + | + | ++ | ++ | ++ |

Notes.
+ Suggests the appropriate disease in the top 100 list of possible diseases.
++ Suggests the appropriate disease in the top 10 list of predictions.
* Does not suggest the appropriate disease in any position of the top 100 list of predictions.

better in this experiment. These two DDX engines succeeded in identifying seven out of ten diseases. ISABEL performed slightly better than DiagnosisPro, as it identified three of the seven diseases among the first ten suggestions. Phenomizer and FindZebra correctly identified nine of the ten diseases. Phenomizer identified five of the nine correct diseases in its top ten of suggestions. FindZebra identified eight of the nine correct diseases in its top ten of suggestions.

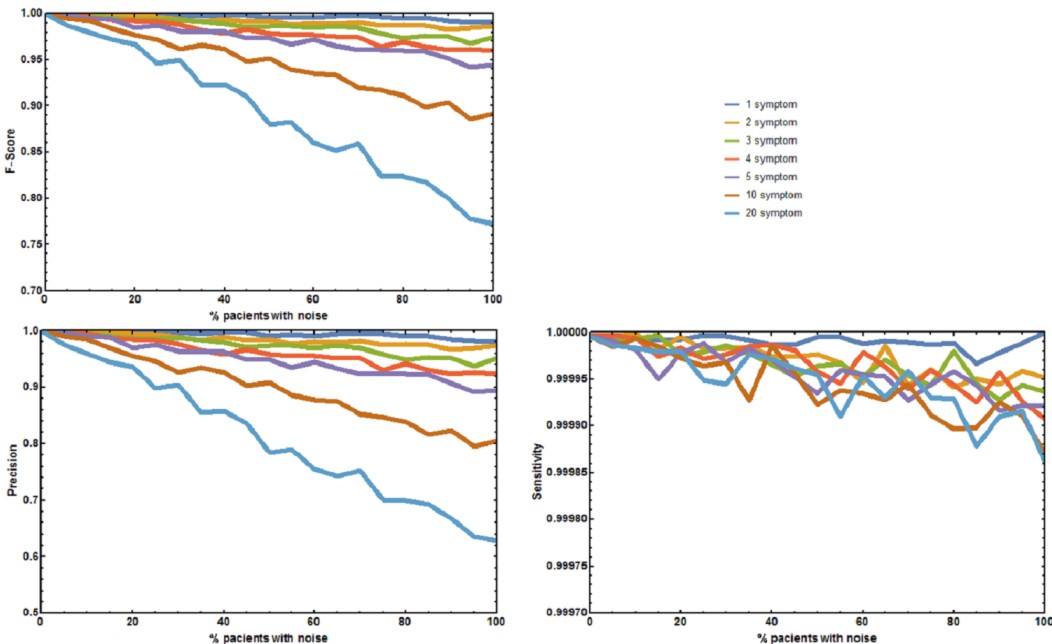

**Figure 2  Joint effect of unreported and unrelated symptoms on the predictive accuracy of Rare Disease Discovery.** (A) Plot of F1-Score as a function of the % of patients with a known rare disease where 1, 2, 3, 4, 5, 10, or 20 symptoms were randomly added or deleted. (B) Plot of Precision as a function of the % of patients with a known rare disease where 1, 2, 3, 4, 5, 10, or 20 symptoms were randomly added or deleted. (C) Plot of Sensitivity as a function of the % of patients with a known rare disease where 1, 2, 3, 4, 5, 10, or 20 symptoms were randomly added or deleted. Without noise, the F1-Score is always 1. The F1-Score decreases as noise (% of patients with deleted symptoms) increases. This is mainly due to a decrease in precision. Sensitivity is always low because the number of false positives is always orders or magnitude smaller than the number of true negatives. In the worst case scenario (20 incorrect symptoms in 100% of the patients), the appropriate disease is contained in the set of diseases with the highest score for more than 80% of the patients.

## Benchmarking the rare disease discovery prototype

Four additional benchmark tests were needed to evaluate the effect of absent and unrelated symptoms on the diagnostic performance of RDD under more realistic, well controlled conditions. The first experiment measured the aggregate effect of absent and unrelated symptoms in predicting the correct disease. The precision $p$, sensitivity $s$, and *F1-Score* of RDD were calculated (Fig. 2). When no symptoms are added or deleted $p$, $s$, and *F1-Score* are always 1, and the correct disease is predicted 100% of the times. As the number of incorrect symptoms increases, $p$ decreases, while $s$ remains approximately constant. Decreases in either $p$ or the *F1-Score* only becomes larger than 5% when the number of symptoms that are randomly added or deleted is equal to or higher than 10 in most of the patients.

The second experiment tested the effect of unreported/absent symptoms in predicting the correct disease. The precision $p$, sensitivity $s$, and *F1-Score* of RDD were calculated (Fig. 3). When no symptoms are deleted $p$, $s$, and *F1-Score* are always 1. The correct disease is predicted 100% of the times. As the number of patients with deleted symptoms increases, $p$ decreases, while $s$ remains approximately constant. Decreases in $p$ or *F-Score* only become
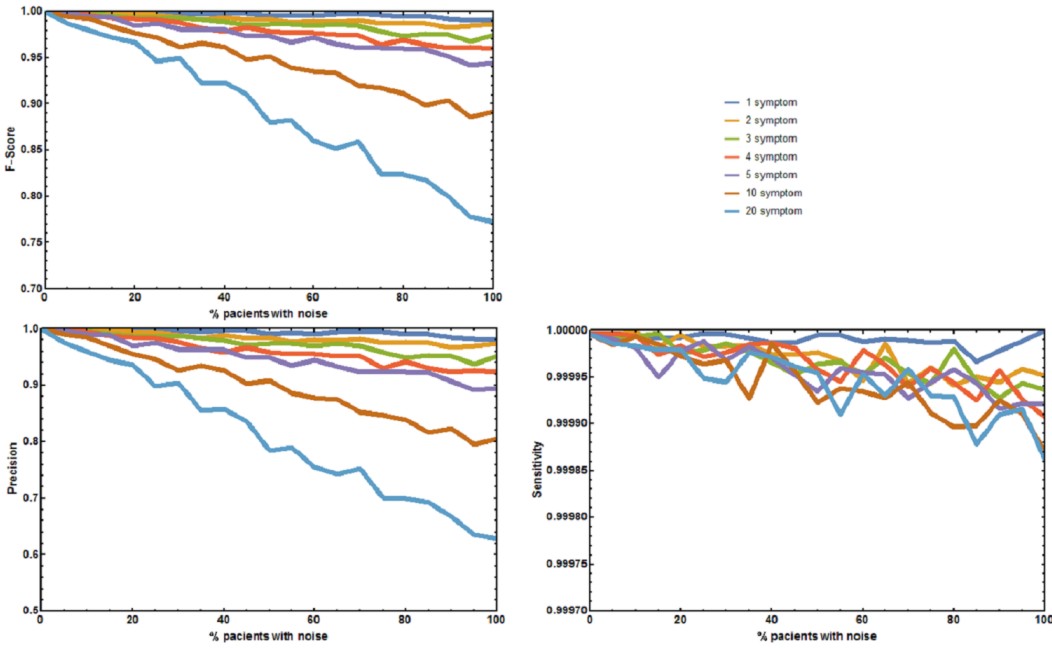

**Figure 3** **Effect of unreported symptoms on the predictive accuracy of Rare Disease Discovery.**
(A) Plot of F1-Score as a function of the % of patients with a known rare disease where 25%, 50%, and
75% of the symptoms were randomly deleted. (B) Plot of Precision as a function of the % of patients
with a known rare disease where 25%, 50%, and 75% of the symptoms were randomly deleted. (C) Plot
of Sensitivity as a function of the % of patients with a known rare disease where 25%, 50%, and 75% of
the symptoms were randomly deleted. Without noise (no deleted symptoms), the F1-Score is always 1.
The F1-Score decreases as noise (% of patients with deleted symptoms) increases. This is mainly due to
a decrease in precision. Sensitivity is always low because the number of false positives is always orders or
magnitude smaller than the number of true negatives. In the worst case scenario (75% deleted symptoms
in 100% of the patients), the appropriate disease is contained in the set of diseases with the highest score
for more than 90% of the patients.

larger than 5% for the sets where 75% of the symptoms are deleted in 50% or more of the
patients.

The two final experiments were used to determine statistical significance of both, the
value of the $DS_i$ score used to rank the diseases and the difference between two $DS_i$ scores.
These experiments estimate that a score $DS_i \geq 0.5$ has a probability lower than 0.0001
of being obtained by choosing a random set of symptoms (see Supporting Methods and
Supporting Figure 3 in Appendix S1). In addition, it also suggests that differences between
$DS_i$ scores lower than 0.01 are significant ($p$-value < 0.001), as long as more than three
symptoms are simultaneously submitted to RDD. If only one symptom is submitted, then
two $DS_i$ scores must differ by more than 0.14 ($p$-value < 0.001). Further details are given
in Supporting Methods and Supporting Table 2 of Appendix S1.

$DS_i$ decreases sharply with noise in all performed experiments; however, even if $DS_i$ is
below the statistically significant level, it can still be used to accurately predict the correct
rare disease, although with a lower confidence (see Supporting Methods and Supporting
Figure 4 in Appendix S1 for details).

## DISCUSSION

### Rare disease discovery

In the vast majority of cases, a definitive and precise diagnostic of a rare disease requires genetic tests. However, in order to direct patients towards the appropriate medical specialists, family doctors need to make a preliminary diagnosis of the potential rare diseases that are consistent with the symptoms observed in the patient. These symptoms are either macroscopic phenotypic observations or clinical parameters from generic biochemical tests. It is at this stage of the diagnosis that rare disease DDX generators are likely to be most useful. Here we presented an approach to create such a DDX generator, Rare Disease Discovery. We implemented that approach as a fast, free, and user-friendly web prototype for initial CAD of patients with suspected rare diseases. We also tested the performance of this prototype in the limited context of the datasets that are available to us.

RDD runs typically take a few seconds, depending on the number of symptoms selected by the user. In the limited conditions under which we could test it, RDD has high precision and sensitivity in our benchmark experiments, suggesting that its diagnostic performance might be robust to situations where not all symptoms have been identified or are directly related with the disease the user is trying to identify. Precision is less robust to these factors than sensitivity, because the number of false positives is always orders or magnitude smaller than the number of true negatives. This makes precision decrease with noise much more sharply than sensitivity. We also show that even when the ranking score $DS_i$ is below significance level, the correct disease is frequently in the set of diseases with the top ten highest $DS_i$. This is also observed in our retrospective studied of 187 previously diagnosed patients. Nevertheless, we remark that testing the performance of the application on much larger, more diverse, anonymized datasets of real patients is needed to validate RDD and its performance. To our knowledge, such datasets are not freely available to the community at present, although they may exist (see below).

Finally, we note that the approach underlying RDD can in principle be extended to any set of diseases. If one has a database associating symptoms to diseases, then one can test the same score function we use and benchmark that score using tests that are similar to the ones performed for RDD, establishing limits of statistical significance for the score function in the context of that database.

### Comparing RDD to similar tools

Interestingly, our illustrative examples of usage suggest that RDD, Phenomizer, and FindZebra have very similar performances while accurately diagnosing rare diseases. RDD and Phenomizer have an equally accurate performance in diagnosing ten out of ten synthetic patients generated from our symptoms database. FindZebra is almost as good, only missing a beta thalassemia diagnosis in one of the ten synthetic patients. This experiment evaluated if the disease used to generate the list of symptoms was included in the list of possible diseases associated to those symptoms. When we randomly select real patients from the RAMEDIS dataset and perform the same experiment, RDD performs slightly better than the other two rare disease DDX tools. While RDD always proposes the disease that was clinically diagnosed to the patient in its top ten of diagnosed diseases,

FindZebra repeats this performance for eight out of ten patients and Phenomizer for five out of ten patients. However, both FindZebra and Phenomizer provide the correct disease in its top 100 list of possibilities for nine out of the ten patients. As expected, RDD, FindZebra, and Phenomizer significantly outperform DDX engines that where designed for CAD of general diseases (ISABEL and DiagnosisPro). We note that these experiments were run using the default settings of all programs. In the case of Phenomizer we also repeated the experiments changing the similarity measure and the multiple testing procedure of the program. However, the results remained qualitatively similar. To be more confident about the comparative performance of the RDD prototype with respect to Phenomizer and FindZebra, a batch study for the one hundred and eighty seven patients from the RAMEDIS dataset would have been desirable. However, a level of access to these tools that would permit automating this study is not available to the general public and this comparison could not be performed. RDD performs fairly well in this dataset, as the clinically diagnosed disease was on the top ten (fifty) list of predictions for more than 60% (80%) of the patients.

We note the qualitatively different approach that these three tools take to ranking the list of possible diseases for a given set of symptoms. Phenomizer takes what we would call a purely statistical approach and calculates the probability that a subset of symptoms could be generated from the complete set of symptom of a disease simply by accident. FindZebra performs a similar analysis for the random occurrence of specific terms in web documents. In contrast, RDD ranks the diseases based on a normalized hamming-like distance between the list of symptoms provided by the user and the list of symptoms from every disease in its database. Internally, RDD establishes the likelihood that a given score is significant or not, informs the user about it, but does not use this significance in the ranking of diseases. We speculate that a meta-server combining RDD, Phenomizer, and FindZebra and providing a consensus diagnostic list would be more accurate than any of the three programs alone. To facilitate this possibility we provide the RDD code and databases as a GitHub project (https://github.com/Wrrzag/DiseaseDiscovery/tree/no_classifiers).

## Limitations

RDD could in the future be developed to become a quick way to assist in initial DDX of rare diseases. This speed comes at the cost of constraining server functionality. For example, admissible symptoms are restricted to those present in the database. This should not be a problem because the data regarding association between rare diseases and symptoms we use comes from ORPHANET. It is well-organized and extensively curated by medical experts. We observed that the improvement in the quality of the ORPHANET annotated dataset leads to an improvement in the predictions made by RDD, as indicated by comparing the benchmark of the server with the ORPHANET data from 2014 to the benchmark of the server with the ORPHANET data from 2015 (see Appendix S1). However, these improvements are small, suggesting that the quality of the ORPHANET disease-symptom annotation is quite high and further improvements to that dataset might not have a significant influence in the performance of RDD.

It is clear that any rare disease that is not included in that database can not be identified by RDD. However, this is also true for all other computer-assisted DDX tools, such as Phenomizer, or FindZebra, which can only identify diseases that are in their respective databases. Overall, the architecture of RDD allows for an easy replacement of the ORPHANET dataset by any other more comprehensive or more adequate dataset that may become available in the future.

An additional important limitation of this study is the size and lack of diversity in the dataset of real patients that we use to evaluate how RDD performs on a real world scenario. This limitation will remain until larger, more diverse datasets of patients are made freely available to the community. We remark that there are projects that have the potential to generate such datasets (e.g., *Choquet & Landais, 2014*; *Koutouzov, 2010*), enabling a more thorough validation of this and other rare disease DDX prototypes. For example, CEMARA reports having data for 235,000 rare disease patients. However, an anonymized version of that data is not readily available for public use. If, or when, such a database becomes available we will use it to further validate and test RDD. In addition we are actively looking for clinical teams that are interested in using the RDD prototype for testing.

## CONCLUSIONS

Rare Disease Discovery is a high performance web prototype for CAD of rare diseases. Its diagnostic performance appears to be robust to situations where not all symptoms have been identified or are directly related with the disease the user is trying to identify. The diagnostic performance of the prototype on a limited set of 187 rare disease patients was good. If this diagnostic performance could be tested and confirmed on larger and more diverse sets of rare disease patients, RDD might potentially become a helpful tool for initial assisted diagnosis of rare disease patients.

## ACKNOWLEDGEMENTS

We thank Drs. Gerard Piñol, Javier Trujillano, and Montse Rue for a critical reading of the paper and helpful suggestions.

### Funding

This work was partially supported by the MEyC under contracts TIN2014-53234-C2-2-R, TIN2011-28689-C02-02 and BFU2010-17704 and by Universitat de Lleida and Departament de Ciències Mèdiques Bàsiques with bridge grants to RA. The authors are members of the research groups 2014-SGR163 and 2014-SGR243, funded by the Generalitat de Catalunya. The funders had no role in study design, data collection and analysis, decision to publish, or preparation of the manuscript.

### Grant Disclosures

The following grant information was disclosed by the authors:
MEyC: TIN2014-53234-C2-2-R, TIN2011-28689-C02-02, BFU2010-17704.

Universitat de Lleida and Departament de Ciències Mèdiques Bàsiques.
Generalitat de Catalunya: 2014-SGR163, 2014-SGR243.

## Competing Interests

The authors declare there are no competing interests.

## Author Contributions

- Rui Alves conceived and designed the experiments, performed the experiments, analyzed the data, contributed reagents/materials/analysis tools, wrote the paper, prepared figures and/or tables, reviewed drafts of the paper.
- Marc Piñol performed the experiments, contributed reagents/materials/analysis tools, wrote the paper, prepared figures and/or tables, reviewed drafts of the paper.
- Jordi Vilaplana and Ivan Teixidó analyzed the data, contributed reagents/materials/analysis tools, reviewed drafts of the paper.
- Joaquim Cruz and Jorge Comas performed the experiments, reviewed drafts of the paper.
- Ester Vilaprinyo analyzed the data, prepared figures and/or tables, reviewed drafts of the paper.
- Albert Sorribas analyzed the data, wrote the paper, prepared figures and/or tables, reviewed drafts of the paper.
- Francesc Solsona conceived and designed the experiments, contributed reagents/materials/analysis tools, prepared figures and/or tables, reviewed drafts of the paper.

## Data Availability

 GitHub: https://github.com/Wrrzag/DiseaseDiscovery/tree/no_classifiers.

## Supplemental Information

Supplemental information for this article can be found online at http://dx.doi.org/10.7717/peerj.2211#supplemental-information.

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
