# Peer review of "Computer-assisted initial diagnosis of rare diseases"

_PeerJ, doi:10.7717/peerj.2211_

## Round 0.1 · original submission · Major Revisions

The Academic Editor has agreed to consider the appeal. Further evaluation of a revised manuscript is requested.

· Appeal

Appeal

Dear Dr. Valencia,

First of all, let me thank you and the reviewers for your work in reviewing this paper. However, we feel that you decision of declining to publish without the opportunity to revise the paper is based on comments that misunderstand the tool and the methodology we use.

Because of that we wish to appeal your decision. Below, we provide the rationale for that appeal by responding to the comments and suggestions made by the reviewers.

Sincerely,

Rui Alves


· · Academic Editor

Reject

Dear Authors,

I really appreciate the originality of your approach. As far as I can see it opens the way to a number of possible applications for the comparison of disease or for the identification of potentially related diseases described in slightly different ways.

Unfortunately, the referees, in particular the second one, describes a number of shortcomings in the design of the study that cannot be solved with a revision, including lack of a sufficiently relevant comparison set and an insufficient number of compared diseases.

I hope you can use the comments and recommendations of the referees, including the useful technical suggestions of the first referee, to improve the work for possible publication.

I am really sorry not to have better news for you today.

Sincerely,

Alfonso Valencia

·

Basic reporting

RDD web application should tell in its main page the version of the platform and the version of ORPHANET used to generate the internal database in use, so users of your DDX generator can properly document the reports or manuscripts which include results from the DDX.

Is the code used for the RDD web application and main calculations available in some public repository, like GitHub? If it is so, you should provide the link to it, as some of the readers could be interested in a local installation.

From the supporting appendix 1, I have understood that it is not possible to automatically generate the MySQL database used by RDD because the input ORPHANET data source is re-curated by you. So, if you provide the code, are you planning to provide periodic MySQL dumps?

Experimental design

As your DDX generator is focused on rare diseases, based on the ORPHANET input data set, something I have missed is a test of how it behaves in the real world: generate a set of synthetic patients based on real diseases which are not in ORPHANET, whose synthetic health records include a partial list of correctly diagnosed symptoms for that disease, plus one or two misdiagnosed ones, which are very frequent in some rare disease. Did you do such test?

As your system takes as input dataset ORPHANET, I have a question about the recorded symptoms in your system. Is your system including only the symptoms from ORPHANET, or is your system also including symptoms from other diseases which are not rare? This question has arisen thinking about the sentence you wrote on line 169 "Symptoms that are absent from the database are not accepted by the server". In a hypothetical scenario where only the list of symptoms related to rare diseases is recorded it could happen that several symptoms which are never related to rare diseases are not used in the diagnosis, giving predictions with better diagnostic scores than it should be.

Validity of the findings

I'm wondering how your algorithm behaves in comorbidity scenarios, where having a rare disease predisposes the patient to have additional diseases, either rare or not, so symptoms from several diseases appear on her/his health record.

Additional comments

Next comments are at the technical level.

Meanwhile I was testing your system (as well as other ones, by the way) I have missed an easier way to provide a list of symptoms, like allowing to put a list of HPO codes.

When the web application shows the results, it would be very useful for the end user to either provide additional information about the disease (other symptoms, full description) or link to a web site where that information is already available, like ORPHANET site. Also, it would be very useful to provide a download link of the results, adding additional columns with technical information like the identifiers of the diseases on one or more disease ontologies or databases.

On the discussion, starting on line 289, I have understood from your explanation that you could not properly evaluate the comparative performance of RDD because it was not possible to run a batch study with the compared systems due they lacked the needed level of automation. My recommendation is that RDD, unlike the compared systems, should provide such automation level, for instance, a REST API, as it would be easier to integrate and use it in the emerging rare disease initiatives and platforms.

Reviewer 2 ·

Basic reporting

No Comments

Experimental design

The method is based on data collected from Orhphanet. This is not described in sufficient details, how it was actually done.

The authors mention that the symptoms data were manipulated, but no details are provided.

Testing of method performance is far from complete. The used data is biased and far too small.

The symptoms are in many cases overlapping for different diseases. Based on Supporting Table 1, only very general symptoms are used. Therefore the given performance figures seem severely overinflated. The method need to be tested with much larger dataset, containing patients both having diseases in those included in here, patients having other disease not included, as well as individuals basically healthy but showing certain symptoms. Only in this way the true performance can be estimated.

Many symptoms appear only on some patients.

Validity of the findings

The applied tests are only anecdotal by nature.

To investigate only 10 diseases or a very limited number of cases from a group of related diseases cannot provide real picture of the performance.

Much larger numbers of cases would be needed for testing. And they should contain also cases not having any of the tested diseases.

For full account of performance reporting it is recommended to follow instructions for related methods in PMID:23169447 The general guidelines presented in the paper apply also in here.

Additional comments

The manuscript describes a method for computer-aided diagnosis of rare diseases. The method is based on disease signs and symptoms collected from Orphanet and on calculation of how many of them match with certain diseases. A score is calculated for that purpose. The method is compared to some related tools, however, there are a number of problems both in the implementation and the comparison.

The title is repetitive and unclear. Should be revised.

According to several recommendations, term "mutation" should be replaced by "variation".

---

## Round 0.2 · Major Revisions

Dear Authors,

First, thanks for solving the technical issues and responding to all the comments of the referees.

The main problem with this paper still persists. As the second referee points out, and you agree in your rebuttal letter, the data set used is too small for a real world application, while it might be sufficient for a first demonstration of the main idea described in the paper.

I agree with the referee, and I am sure you will also agree, this type of system are prototypes that are still too far from the real world and they are not at this point ready for their use by clinicians. For that, this or any other system, will need to be developed to a completely different level including a full testing with a much larger dataset.

Still, I value the idea and the implementation that I think deserve publication since they may inspire others that may have access to larger data sets.

In this situation, I return the paper to you. We will publish the paper only if you formulate the text to celery indicate the limitations imposed by the datasets used, make clear that this is a prototype that will have to be recast with new data, and avoid expression that could make anyone think that your are proposing this as a system ready to be used by clinicians in real practice.

I will of course understand that you may prefer to submit the work to some other journal instead.

Sincerely,

Alfonso Valencia

·

Basic reporting

As I already recommended you, end users (i.e. clinicians) of RDD web application which use it may need to know the version of ORPHANET used to train the platform's database, as it can be relevant for their own reports.

Experimental design

No Comments

Validity of the findings

No Comments

Additional comments

I have found several issues in the manuscript, which are enumerated below (referred to the line number from the reviewing PDF):

* BACKGROUND section, line 72. The sentence, "In addition, as can be seen ..." should be "In addition, as it can be seen ...".
* BACKGROUND section, line 91. The sentence "... as is the case for ... " sound a bit strange in the context of the whole sentence.
* BACKGROUND section, line 95. Repeated 'be', "... it can be also be used for ...".
* 'Data Source & Software' subsection, line 113. "... diseases, with its associated symptoms was", maybe a comma is needed between 'symptoms' and 'was', taking into account the whole sentence.
* 'Illustrative examples of RDD usage' subsection, line 185. The start of the sentence resembles more the style of a book or a blog entry's sentence, instead of a manuscript's one.
* 'Retrospective Study of Previously Diagnosed Rare Disease Patients' subsection, line 209. Sentence "In 60% (80%) of these patients, the clinically diagnosed disease was on the top ten (fifty) list of predictions" is very unclear for me, as I don't get what it is being described with the percentage and number in parentheses.
* DISCUSSION section, lines 314 to 317. 'orphanet' should be written as 'ORPHANET' in order to keep consistency with the whole manuscript.
* DISCUSSION section, line 323. "In the meantime Wwe plan ...", a typo writing 'we'.

Reviewer 2 ·

Basic reporting

The authors have aimed at improving their method, however, basically nothing has been done to mitigate the problems indicated in the first evaluation.

Experimental design

The method cannot work in its intended purpose. Clinicians cannot trust on it. The authors are limited by existing data, that is the situation in all research and method development. The problem in here is that for many diseases the signs and symptoms are so general that they would apply to many other, including common diseases, and even healthy individuals.
To test this kind of method much larger set of cases would be needed. It is not easy task to obtain those cases, but there are thousands, likely tens of thousands, of cases reported in literature.
When a method is intended for clinical use, as the authors claim, there are very strict requirements for the performance and testing of such tools. What has been done is nowhere near those demands.

Validity of the findings

Due to far too small sample size it is practically impossible to say anything about the quality of the method. Further, the used signs and symptoms are not sufficient for diagnosis of many diseases, whether rare or common.

---

## Round 0.3 · accepted · Accept

Dear Authors,

Thanks very much for the extensive revision of the paper and for addressing the key concerns raised by the referees.

I am convinced that this "conservative" version will make clear the potential and limitations of the current datasets and approaches without making unnecessary and potentially damaging over claims.

Sincerely,

Alfonso Valencia